# Soil Moisture Data Assimilation in MISDc for Improved Hydrological Simulation in Upper Huai River Basin, China

Zhenzhou Ding [1,2], Haishen Lü [1,2,*], Naveed Ahmed [1,2], Yonghua Zhu [1,2], Qiqi Gou [1,2], Xiaoyi Wang [1,2], En Liu [1,2], Haiting Xu [1,2], Ying Pan [1,2] and Mingyue Sun [1,2]

1   State Key Laboratory of Hydrology-Water Resources and Hydraulic
    Engineering, College of Hydrology and Water Resources, Hohai University, Nanjing 210098, China
2   College of Hydrology and Water Resources, HoHai University, Nanjing 210098, China
*   Correspondence: lvhaishen@hhu.edu.cn; Tel.: +86-17361875862

**Abstract:** In recent years, flash floods have become increasingly serious. Improving the runoff simulation and forecasting ability of hydrological models is urgent. Therefore, data assimilation (DA) methods have become an important tool. Many studies have shown that the assimilation of remotely sensed soil moisture (SM) data could help improve the simulation and forecasting capability of hydrological models. Still, very few studies have attempted to assimilate SM data from land surface process models into hydrological models to improve model simulation and forecasting accuracy. Therefore, in this study, we used the ensemble Kalman filter (EnKF) to assimilate the China Land Data Assimilation System (CLDAS) SM product into the MISDc model. We also corrected the CLDAS SM and assimilated the corrected SM data into the hydrological model. In addition, the effects of the 5th and 95th percentiles of flow were evaluated to see how SM DA affected low and high flows, respectively. Additionally, we tried to find an appropriate size for the number of ensemble members of the EnKF for this study. The results showed that the EnKF SM DA improved the runoff simulation ability of the hydrological model, especially for the high flows of the model; however, the simulation for the low flows deteriorated. In general, SM DA positively affected the ability of the MISDc model runoff simulation.

**Keywords:** data assimilation (DA); soil moisture (SM); CLDAS; the MISDc model; direct insertion (DI); ensemble Kalman filter (EnKF)

## 1. Introduction

In recent years, with global climate change, the problem of flash floods has become more and more serious, and the problem of flood forecasting has become a key issue. The simulation and forecasting of floods in small- and medium-sized river basins (usually <3000 km$^2$), especially, has been a global problem. Many hydrological models have been developed in the past decade to simulate and predict flood hazards. Due to the simple structure and small computational effort of lumped hydrological models (which generalize the watershed as a whole, ignoring the influence of the local unevenness of geology, geomorphology, soil, vegetation and other elements on the hydrological cycle within the watershed), they are often used to forecast floods in small- and medium-sized river basins, and their forecast accuracy is no less than that of some distributed hydrological models [1–3]. Uncertainties in a hydrological model's structure and parameters [4], as well as the quality of the data used to drive the hydrological model [5–8], can lead to errors in runoff simulation and the misrepresentation of the real flood processes. Therefore, in recent years, many scientists have proposed using data assimilation (DA) methods to correct the process of hydrological model runoff simulations in anticipation of improving the forecast accuracy of hydrological models [9,10].

DA is a key technique for modeling, as it is a way to evaluate and validate prediction models, and it has gradually come to the forefront of research on ecohydrological processes

and remote sensing inversion [11–14]. DA can consider the uncertainties of model structure and model input data, and it can be used to make real-time corrections of hydrological state variables (such as runoff and soil moisture (SM)) combined with observation data in order to reduce the uncertainties of state variables and improve the prediction accuracy of hydrological models. As a common DA method, the ensemble Kalman filter (EnKF) has been widely used in SM DA [15,16]. The EnKF is a Monte-Carlo simulation based on the Kalman filter (KF), which uses a set of state variables and observations superimposed with noise perturbations to generate a set of analytical values (ensemble) of a system state field, then assumes that the ensemble mean is true, estimates the error of each ensemble member, obtains the error covariance matrix of the state field, and then uses the new observations to update the error covariance [17]. The EnKF method has received much attention in the field of hydrology because of its advantages in dealing with nonlinear problems in the context of hydrological models.

SM is a key variable that regulates surface water and groundwater interactions as well as energy exchanges between land and atmosphere [18]. In particular, SM measurements contain information about the pre-storm hydrological state and storm rainfall estimates, which are critical for accurate runoff simulations. Therefore, the assimilation of SM into hydrological models has become a viable approach to improve model runoff simulations [19]. Brocca, et al. [20] showed that pre-event SM conditions impacted runoff simulations, noting that the runoff characteristics of a watershed (runoff depth and peak flow) were highly correlated with pre-event SM, as assessed by the MISDc model. Therefore, SM can be assimilated into a hydrological model to improve the model's forecast accuracy. Due to the spatial heterogeneity of soil texture, the spatial distribution of SM significantly varies from location to location. As a result, it is difficult to obtain accurate regional or even global SM by only relying on measurements from ground stations. Therefore, remote sensing observations of SM and estimates acquired from the land surface process have become necessary [21]. Large-scale SM data can be conveniently obtained through remote sensing monitoring. However, remote sensing monitoring also has shortcomings in time and space resolution, and errors in remote sensing data are generally larger than those of ground station observation data. In addition, remote sensing monitoring can only provide surface measurement, but hydrological simulations are concerned with SM at both the surface layer and the root zone layer. Moreover, many studies have demonstrated that assimilated remote sensing SM data can improve the accuracy of SM estimation and runoff simulation in land surface process and hydrological models. For example, De Santis, et al. [22] assimilated SM data from the European Space Agency (ESA) Climate Change Initiative (CCI) into the MISDc-2L model in more than 700 small- and medium-sized river basins in Europe and found that the DA method improved the simulation accuracy of the hydrological model in many basins; Gavahi, et al. [23] assimilated SM data from SMOS and evapotranspiration data from MODIS into the VIC model and found that the DA algorithm improved the model simulation not only for surface SM (SSM) but also for root zone SM (RZSM), and they used the SM data obtained by the DA algorithm for drought studies.

However, very few studies have assimilated SM data calculated by a land surface process model into a hydrological model in DA experiments to see whether the land surface process model positively affected the hydrological model forecasts. The coupling of land surface process models with hydrological models, especially with grid hydrological models, is relatively easy. In flash flood forecasting, lumped hydrological models are most often used. The structure of a lumped hydrological model is different from that of a land surface process model, so how to couple land surface and lumped hydrological models and how to use the simulation results of a land surface model to improve the simulation accuracy of a lumped hydrological model are questions worth considering. In this paper, we assimilated land surface process SM data into a lumped hydrological model (MISDc) as an example and propose the idea that lumped hydrological models can be used for hydrological flood forecasting studies. There are many land surface process data products that can be generated (e.g., SM data, evapotranspiration). If these products can be used in

lumped hydrological models, could they have positive effects on hydrological modelling studies? The influence of land surface processes on hydrological processes was considered. The fact that the studied land surface process model had RZSM data and that the data were updated in real time facilitated the SM DA experiments.

Although land surface process models can, under certain circumstances, make accurate estimates of SM, they also contains many uncertainties [24]. For example, (1) the quality and quantity of the data used to drive a model has a significant influence on the quality of its final outputs [5–8]; (2) parameter estimation errors of a model are inevitable, which may lead to large errors at the model input site [4]; and (3) the simple representation of real processes in nature by land surface process models and hydrological models lead to assumptions and simplification errors, which lead to problems in the simulation of complex, realistic conditions [25]. However, the SM data from the land surface process model can be bias -corrected to obtain a more accurate set of SM data before assimilation studies of SM data are conducted. Wang, et al. [26] used a back-propagation neural network (BPNN) to train SM data at 0–10 cm of the China Land Data Assimilation System (CLDAS) product and 0–10 cm of in -situ observations in the Huai River Basin. A new set of SM data (CLDAS-BPNN SSM) was obtained. The results showed that the BPNN method could reduce the bias of the original CLDAS results without destroying the temporal correlation between the original CLDAS results and the in-situ observations.

The main DA method used in this paper was the ensemble Kalman filter (EnKF) method, and the direct insertion (DI) method was compared with it. The hydrological model used in this paper was the event-based rainfall–runoff (RR) "Modello Idrologico Semi-Distribuito in continuo" (MISDc) model. This model was first proposed by Brocca, et al. [27] and then improved by Brocca, et al. [28], and the improved version was used in this paper. Although the MISDc model has a simple structure, its simulation accuracy of flood processes is adequate [24,29]. In addition, the single-layer version of the MISDc model can directly assimilate SM data of a corresponding soil layer once the soil layer thickness is determined, which was of great convenience to our research. The MISDc model has been widely applied in Italy and other European countries, as well as in research on remote sensing SM data assimilation [20,22,24,29–33]. Therefore, this paper used the MISDc model for SM DA experiments. The study was carried out in three sub-basins of the Huai River Basin in China, ranging in area from 200 to 3000 km$^2$. The SM data used in this paper were from a CLDAS SM product published by the China Meteorological Administration (http://data.cma.cn (accessed on 10 September 2022)). Its data accuracy in China is superior to competing SM products [34]. In addition, the cumulative distribution function (CDF) matching method [35] was applied to match the CLDAS-BPNN SSM data with the CLDAS data to calculate a five-layer time series of CDF SM data.

In this work, due to the uncertainties of the input forcing data of the MISDc model, the representation of SM by the model was found to be poor, which led to poor runoff simulation. Therefore, we intended to use DA technology to improve the ability of the MISDc model to simulate runoff in the Huai River Basin of China. Since the accuracy of CLDAS SM data in the Chinese region is higher than other products of the same type and because the data contain precious information on SM variability, we attempted to assimilate CLDAS SM data into the MISDc model to answer the following questions:

1.  Which layer of CLDAS SM data is most suitable for DA experiments in the MISDc model? (Section 2.6);
2.  To what extent does the assimilation of CLDAS SM products improve the runoff simulation capability of the model? What is the effect of SM bias correction (i.e., using CLDAS-BPNN SSM data) on the results? (Section 3.3);
3.  What are the implications of using the EnKF method for high and low-flow simulations in the hydrological model? (Section 3.4);
4.  What is the effect of different ensemble numbers on the results of the EnKF method for the hydrological model runoff simulations? (Section 3.2).

The structure of this paper is as follows: Section 2 introduces the study area and the data we used, and also presents the specific operation process of the hydrological model (MISDc), the CDF, and the EnKF method, as well as the performance indexes used in this paper. Section 3 introduces the results of this paper. Section 4 discusses the results of this paper. Section 5 is a summary of this paper.

## 2. Materials and Methods

### 2.1. Study Area

The Huai River Basin, which spans the Henan, Anhui, Jiangsu and Shandong provinces, plays an extremely important role in Chinese economic and social development. The arable land area of the basin is about 130,000 km$^2$ (accounting for about 12% of the total arable land area of the country), the grain output accounts for about 1/6 of the total output of the country, and the commodity grain supply accounts for about 1/4 of the total output of the country [36]. As a result, this basin plays a pivotal role in the national food security system.

The Huai River Basin is located between 111°55' E to 121°25' E and 30°55' N to 36°36' N, covering an area of 274,657 km$^2$. It is located in the transitional zone of climate between the north and south of China and belongs to the semi-humid monsoon climate zone of a warm temperate zone. Its annual average temperature ranges from 11 °C to 16 °C, increasing from north to south and from coastal to inland. The highest monthly average temperature in July is about 27 °C, and the lowest monthly average temperature in January is about 0 °C. The relative humidity in the basin is relatively high, with an average of 66%~81% for many years. In summer, it is generally more than 80%, and in winter, it is about 65%. The annual sunshine hours range from 1990 to 2650 h. The average annual precipitation is about 895 mm, and the average annual runoff depth is 221 mm. The annual average surface evaporation is 1060 mm.

To illustrate the generalizability of the MISDc model for SM DA on the Huai River basin, three small- and medium-sized river basins in the Huai River basin were randomly selected. The Wangwuqiao (WWQ), Dapoling (DPL) and Changtaiguan (CTG) sub-basins of the Wangjiaba Basin in the southwest of the Huai River Basin were selected as per the study area. The specific locations of the three basins are shown in Figure 1. The areas of the WWQ, DPL, and CTG basins are 200, 1640 and 3090 km$^2$, respectively. As can be seen from Figure 1c, the WWQ Basin is relatively flat and belongs to a plain landform, with an elevation ranging from 51 to 84 m with little difference. As can be seen from Figure 1b, the DPL basin is a sub-basin of the CTG basin, which has a high altitude, with the highest areas reaching 1110 m in elevation and a net elevation difference of more than 1000 m.

### 2.2. Observed Discharge Data

The observed discharge data of the three hydrological stations (Wangwuqiao station, Dapoling station and Changtaiguan station) used in this paper were from the daily mean flow tables of the Hydrological Yearbook. The time series of the data was from 1 January 2010 to 31 December 2017, of which 1 January 2010 to 31 December 2014 was used for the calibration of the MISDc model and 1 January 2015–31 December 2017 was used for validating.

### 2.3. China Land Data Assimilation System Product

#### 2.3.1. Forcing Data

The rainfall and temperature data were from the CLDAS atmospheric forcing product V2.0, which covers the Asian region (0–65° N, 60–160° E), with a spatial resolution of 0.0625° × 0.0625° and a temporal resolution of 1 h. Its quality in China was evaluated using more than 2400 national automatic station time observations.

The rainfall product was formed by interpolating satellite precipitation products (FY2/EMSIP precipitation products) to analyze grid points outside the Chinese region and by interpolating fused rainfall products to analyze grid points within the Chinese territory.

The correlation coefficient (*R*) of hourly rainfall data was calculated as 0.72, the *RMSE* was 0.94 mm/h, and the deviation was −0.004 mm/h. The unit of the product is mm/h.

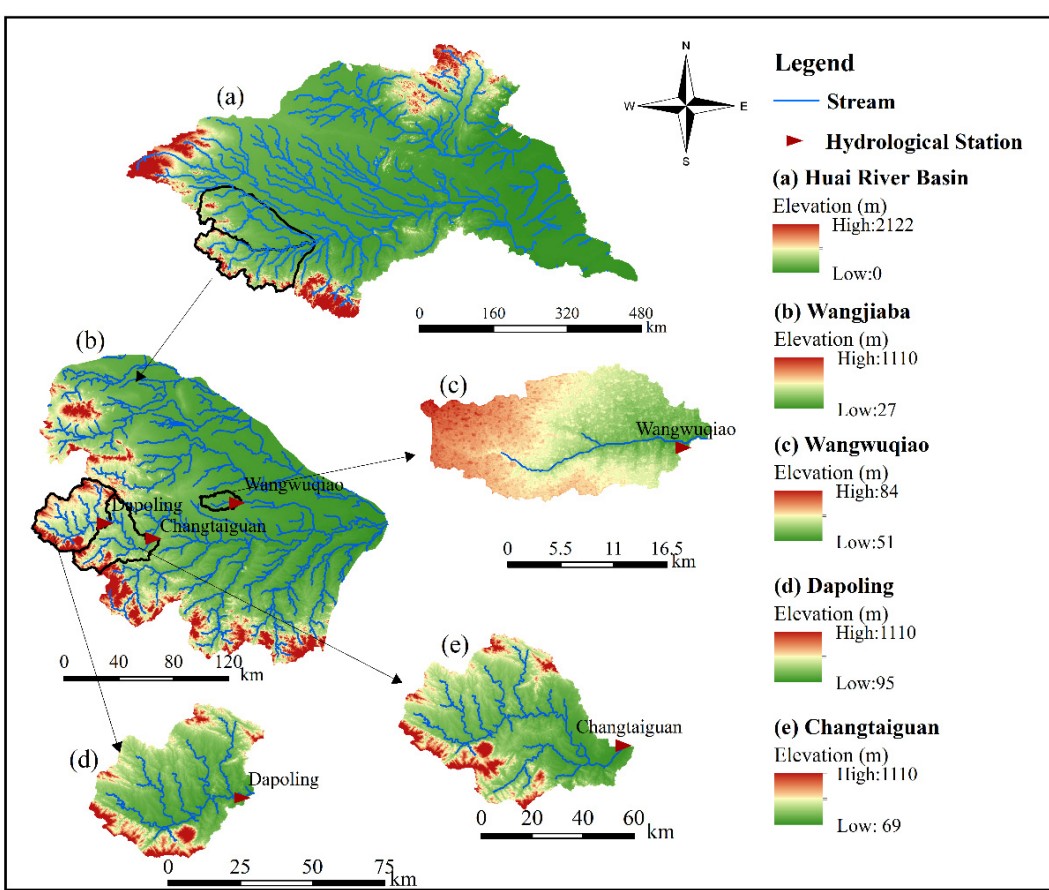

**Figure 1.** Study area. (**a**) Huai River Basin; (**b**) Wangjiaba Basin; (**c**) Wangwuqiao (WWQ) basin; (**d**) Dapoling (DPL) basin and (**e**) Changtaiguan (CTG) basin.

The temperature product was formed by using the ECMWF numerical analysis/forecast product as the background field. The part within the Chinese region was formed by fusing the ground-based automatic station observation data using topographic adjustment and multiple grid variation techniques (STMAS). The part outside the Chinese territory was created with topographic adjustment, the variable diagnosis of the background field, and interpolation of the analysis grid points. The temperature product's correlation coefficient (*R*) was 0.97, the *RMSE* was 0.88 K, and the deviation was −0.13 K. The unit of the product is K. The average CLDAS values of all grids in the three basins were used as rainfall and temperature data of the corresponding study area.

2.3.2. Soil Moisture Data

The SM data from the CLDAS SM product were based on the multi-model averaging of three different land surface models (i.e., CLM3.5, CoLM, and Noah-MP) and generated by the China Meteorological Administration within an East-Asian domain that covers the Asian region (0–65° N, 60–160° E), with a spatial resolution of 0.0625° × 0.0625° and a temporal resolution of 1 h.

The SM analysis product contains five soil layers between 0 and 5, 0 and 10, 10 and 40, 40 and 100, and 100 and 200 cm. The product was in good agreement with actual observations on the ground. The national average correlation coefficient (*R*) was 0.89, the *RMSE* was 0.02 m$^3$/m$^3$, and the deviation was 0.01 m$^3$/m$^3$. The unit of the product is m$^3$/m$^3$. The average CLDAS SM values of all grids in the three basins were used as SM data of the corresponding study area.

Because of the systematic deviation caused by model parameters and structural errors of CLDAS SM products, we needed to conduct deviation correction of the CLDAS SM data before conducting our SM DA experiments. Wang, et al. [26] used a back propagation neural network (BPNN) model to map the relationship between in situ SM observations and each corresponding CLDAS grid at the original spatial resolution in the Huai River Basin, China. Based on his study, the CLDAS-BPNN SSM data were generated. According to his work, the CLDAS-BPNN data can reduce the bias in the original CLDAS results without damaging the temporal correlation versus in situ observations, and the CLDAS-BPNN data showed better results than the original CLDAS data. Based on these positive results, the CLDAS-BPNN product can be used for data assimilation [26].

It is worth noting that the CLDAS-BPNN SMM data corresponded to the CLDAS SM data at 0–10 cm, and the SM data correction for the other layers of the CLDAS is described in Section 2.5.

### 2.4. Hydrological Model

The hydrological model used in this paper was the MISDc model ("Modello Idrologico Semi-Distribuito in continuo"). The MISDc model is a single-layer model, and it consists of two main components, i.e., a soil water balance (SWB) model and a semi-distributed event-based rainfall–runoff (RR) model. The SWB model simulates the temporal patterns of SM and sets initial conditions for the second component (the RR model, used for flood process simulation). The two models are coupled through a simple linear relationship between the saturation ($W(t)/W_{max}$) and the soil retention parameter ($S$) of the soil retention service curve number (SCS-CN) method [20].

The SWB model considers the temporal evolution of soil water in a single soil layer, and the soil water balance equation is the main equation, that includes infiltration, drainage and evapotranspiration. The Green–Ampt equation expresses the infiltration, the drainage is represented by a nonlinear gravity equation [37], and the potential evapotranspiration ($ET_p(t)$) is expressed according to the modified Blaney–Criddle equation [38]. The RR model uses the SCS-CN method to estimate loss and geomorphic instantaneous unit process lines (IUH) and linear reservoir IUH to route excess rainfall in catchments and in areas directly discharging into the main channel, respectively [39].

The MISDc model contains eight parameters and is characterized by a low computational effort that is very attractive for hydrological practice. The details of the parameters are listed in Table 1. For a detailed description of the MISDc model, the reader is referred to the work of Brocca, et al. [30] and Brocca, et al. [20].

**Table 1.** Description of MISDc model calibration parameters, units and ranges.

| Model Component | Parameter | Description | Unit | Range |
|---|---|---|---|---|
| SWB | $W_{max}$ | Maximum water storage of the soil layer | *mm* | 100–1000 |
| | $K_s$ | Saturated hydraulic conductivity | *mm/h* | 0.01–20 |
| | $m$ | Drainage exponent | - | 5.0–60 |
| | $Nu$ | Fraction of drainage versus interflow | - | 0–1.0 |
| | $b$ | Correction coefficient for the potential evapotranspiration | - | 0.4–2.0 |
| RR | $\eta$ | Lag–area relationship parameter | - | 0.5–6.5 |
| | $\lambda$ | Initial abstraction coefficient | - | 0.0001–0.2 |
| | $a$ | Relationship between modelled SM and the *S* of the SCS-CN method | - | 1.0–5.0 |

### 2.5. Cumulative Distribution Function

The cumulative distribution function (CDF) was first proposed by [35]. We used this method to relate the 0–10 cm layer of CLDAS SM data to other layers one by one and then extrapolate the SM data of different layers from CLDAS-BPNN SSM data. Figure 2 shows a

concrete representation of this process. The central content is the same $cdf$ value between the matching data, i.e.:

$$cdf(X_{CLDAS-BPNN}) = cdf(X_{CLDAS\_Li}) \tag{1}$$

where $X_{CLDAS-BPNN}$ stands for the SM data from the CLDAS-BPNN SSM (the SM data of 0–10 cm) and $X_{CLDAS\_Li}$ stands for the other layers SM (besides 0–10 cm) from CLDAS.

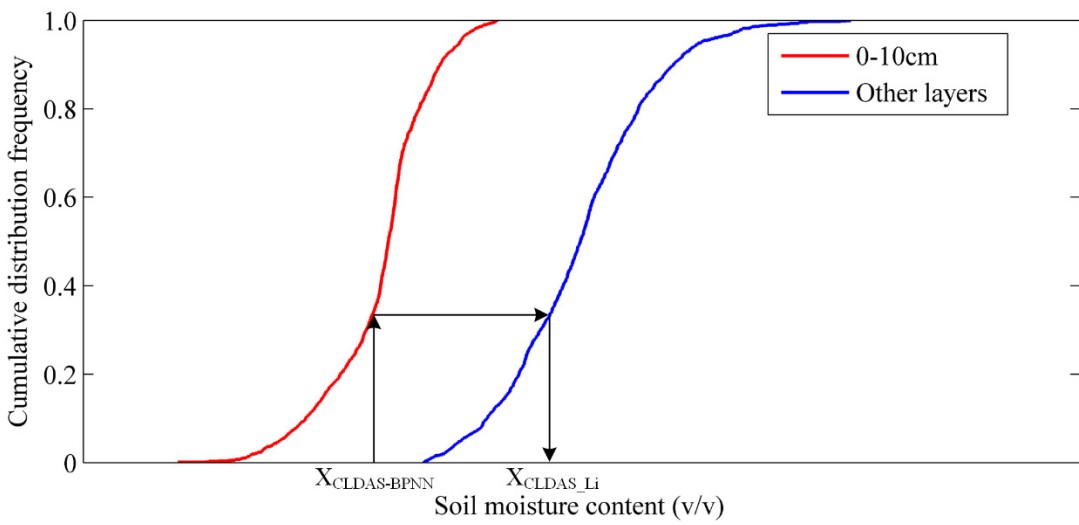

**Figure 2.** A graph showing how the CDF of CLDAS-BPNN SSM was adjusted into that of SM at other layers.

So, based on the CLDAS-BPNN SSM data, the SM data from the five CLDAS layers were reanalyzed and processed with the CDF method (Equation (1)) to generate the corresponding BPNN-05, BPNN-40, BPNN-100 and BPNN-200 data for subsequent DA experiments. For the convenience of writing, the CLDAS-BPNN SSM data are recorded as BPNN-10 in a uniform format.

*2.6. Determination of the Thickness of the Model Soil Layer*

Since the CLDAS SM data used in this paper represented soil water content and soil water storage was required to conduct the DA experiment, the magnitude of the soil thickness ($L$) simulated by the model needed to be determined when performing the transformation of these two variables. The determination was conducted by calculating the correlation between the soil water storage obtained from the model simulation and the soil water storage of CLDAS at different depths ($z$), with $z$ corresponding to the maximum value of the correlation was the thickness of the soil layer of the MISDc model in the basin [24,32].

In this paper, SM information for any soil depth between 0 and 200 cm, $\theta_{CLDAS}$, was obtained via the weighed mean of the soil moisture provided by the related layer, according to:

$$
\begin{aligned}
\theta_{CLDAS} &= \theta_1 & &: z \leq 5\,\text{cm} \\
\theta_{CLDAS} &= \theta_2 & &: 5 < z \leq 10\,\text{cm} \\
\theta_{CLDAS} &= \frac{10\theta_2 + (z-10)\theta_3}{z} & &: 10 < z \leq 40\,\text{cm} \\
\theta_{CLDAS} &= \frac{10\theta_2 + 30\theta_3 + (z-40)\theta_4}{z} & &: 40 < z \leq 100\,\text{cm} \\
\theta_{CLDAS} &= \frac{10\theta_2 + 30\theta_3 + 60\theta_4 + (z-100)\theta_5}{z} & &: 100 < z \leq 200\,\text{cm}
\end{aligned}
\tag{2}
$$

where $\theta_i$ ($i$ = 1, 2, 3, 4, 5) is the SM for each of the five layers of the CLDAS product and $z$ is the parameter representing the depth. The SM data obtained from CDF calculations at different depths were processed in the same way.

Since the time series of the DA experiment was from 31 March 2015 to 31 March 2018, the data from the same time series were to determine the thickness of the soil layer for

the model. We set $z$ to vary from 0 to 200 cm, calculated a series of soil water contents according to Equation (2), and correlated them with the model output soil water content. Figure 3 shows the trend of the correlation coefficients between the soil water storage at different depths of CLDAS and the soil water storage obtained from the model simulation. From Figure 3, we can see that the value of $R$ calculated for these three basins reached its maximum value near $z = 100$ cm. As a result, integrated soil water storage from 0 to 100 cm was used to form the data for the assimilation experiment.

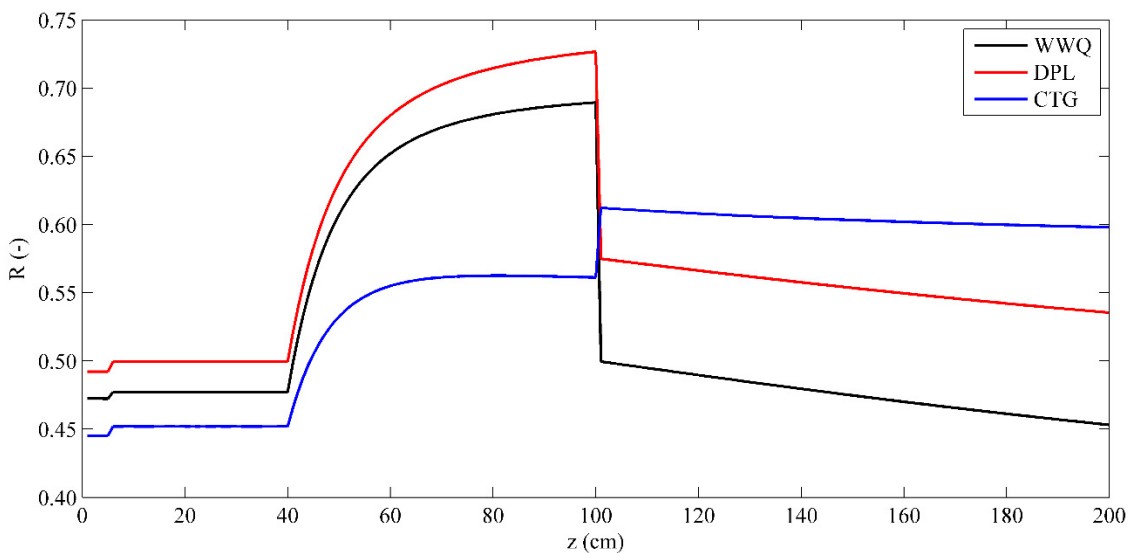

**Figure 3.** The correlation coefficient (R) between soil water storage of MISDc and CLDAS at different depths for the three basins.

### 2.7. Ensemble Kalman Filter

The ensemble Kalman filter (EnKF), as a sequential data assimilation method, was proposed by Evensen [9] and has been widely used in various fields. Since most land surface processes are moderately nonlinear and the EnKF can deal with such systems with high reliability, it has been widely recognized and applied to land surface modeling. In particular, the EnKF has the following advantages: (1) the assimilation system is relatively easy to implement; (2) for discontinuous and nonlinear models, good assimilation results can also be achieved; and (3) based on the Kalman filter, it overcomes the weakness that model operators need to be linearized.

The members ($N$ members) of the system synthesis updated at time $k$ ($X_{i,k}^{f}, i = 1, 2, \ldots, N$) are propagated through the state transfer function:

$$X_{i,k+1}^{f} = f(X_{i,k}^{a}, u_{i,k}, v_{i,k}) \tag{3}$$

where superscripts $a$ and $f$ represent the analytical process and the forecast process, respectively. $X_{i,k+1}$ is the value of the $ith$ state variable at time $k + 1$; $u_{i,k}$ represents the perturbed forcing data and $v_{i,k}$ is white Gaussian noise with mean equal to zero and variance equal to $Q_k$.

The main equation for EnKF implementation is as follows:

$$X_{i,k+1}^{a} = X_{i,k+1}^{f} + K_{k+1}\left[Z_{k+1} - H(X_{i,k+1}^{f}) + \eta_{i,k+1}\right] \tag{4}$$

where $Z$ is the observed value, $H$ is the observation operator, $\eta_{i,k+1}$ is white Gaussian noise with a mean equal to zero and variance equal to $R_{k+1}$, and $K_{k+1}$ is the Kalman gain matrix [40]:

$$K_{k+1} = P_{k+1}^{f}H^{T}(HP_{k+1}^{f}H^{T} + R_{k+1})^{-1} \tag{5}$$

In Equation (5), $P_{k+1}^f$ is the priori information error variance prediction matrix at the moment $k + 1$:

$$P_{k+1}^f = \frac{1}{N-1} \sum_{i=1}^{N} (X_{i,k+1}^f - \overline{X_{k+1}^f})(X_{i,k+1}^f - \overline{X_{k+1}^f})^T \tag{6}$$

where $N$ is the number of ensemble members, which is discussed in Section 3.2.

Figure 4 shows the application of the EnKF method and the DI method on the MISDc model.

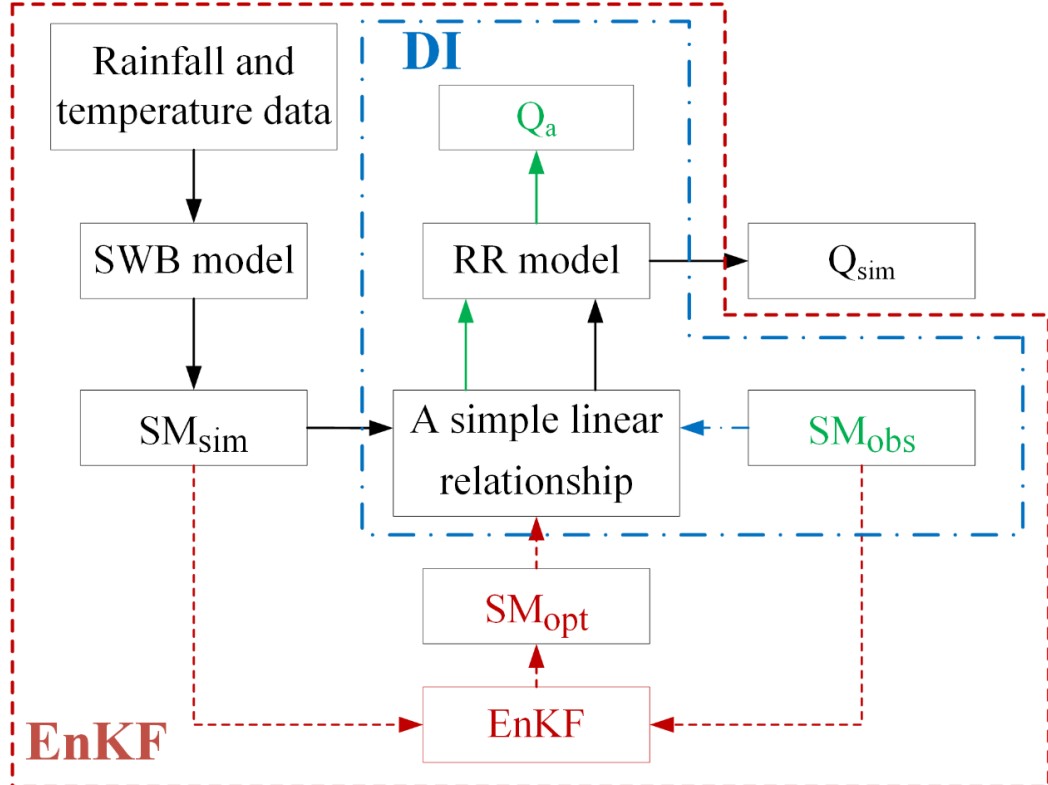

**Figure 4.** Flowchart of DA for soil water storage using the DI method (blue) and the EnKF method (red) coupled with the MISDc model; the green parts are common between the two methods.

*2.8. Performance Indexes*

Three indexes were used to describe the model performance during the calibration and validation periods. The first one was the Nash efficiency coefficient, $NS$ [41]:

$$NS = 1 - \frac{\sum\limits_{t=1}^{n} (Q_{obs} - Q_{sim})^2}{\sum\limits_{t=1}^{n} (Q_{obs} - \overline{Q_{obs}})^2} \tag{7}$$

where $n$ is the period of analysis, $Q_{obs}$ is the observed discharge, $Q_{sim}$ is the discharge simulated by the model and $\overline{Q_{obs}}$ is the mean discharge during the period $n$.

The second one was $ANSE$ (Equation (8)), which evolved from $NS$ and is also used to describe high flow conditions [42]. The third one was the correlation coefficient, $R$, which was used to reflect the degree of fit between $Q_{obs}$ and $Q_{sim}$.

$$ANSE = 1 - \frac{\sum\limits_{t=1}^{n} (Q_{obs} + \overline{Q_{obs}})(Q_{sim} + Q_{obs})}{\sum\limits_{t=1}^{n} (Q_{obs} + \overline{Q_{obs}})(\overline{Q_{obs}} + Q_{obs})} \tag{8}$$

In order to evaluate the improvement of the model runoff simulation via data assimilation, we used the Normalized Root Mean Squared Error ($NRMSE$) to qualitatively describe it [43]:

$$NRMSE = \frac{\frac{1}{N}\sum\limits_{i=1}^{N}\sqrt{\frac{1}{n}\sum\limits_{t=1}^{n}\left(Q_a^i(t)-Q_{obs}(t)\right)^2}}{\frac{1}{N}\sum\limits_{i=1}^{N}\sqrt{\frac{1}{n}\sum\limits_{t=1}^{n}\left(Q_{sim}^i(t)-Q_{obs}(t)\right)^2}} \qquad (9)$$

where $Q_a$ is the discharge obtained by using the two data assimilation methods. At the same time, to separately evaluate the impact of data assimilation technology on high and low flows, we used the $ANSE$ index for the high flows and the $NS_{(logQ)}$ index for the low flows. $NS_{(logQ)}$ is the $NS$ of the logarithm of the flow [33]:

$$NS_{(\log Q)} = 1 - \frac{\sum\limits_{t=1}^{n}\left[\log(Q_{sim}+\varepsilon)-\log(Q_{obs}+\varepsilon)\right]^2}{\sum\limits_{t=1}^{n}\left[\log(Q_{obs}+\varepsilon)-\log(\overline{Q_{obs}}+\varepsilon)\right]^2} \qquad (10)$$

where $\varepsilon$ is a very small number to ensure that the value of $\log(\cdot)$ is not zero.

For all of the above evaluation indicators, the closer the value of $NS$ and $R$ is to one, the better the simulation results will be. An $NRMSE$ greater than one indicates the deterioration of data assimilation results, while an $NRMSE$ of less than one indicates relative improvement. For the difference between simulated values of the high and low flow before and after data assimilation, the results of $\Delta ANSE$ and $\Delta NS_{(logQ)}$ are worse if less than zero and improved if greater than zero.

The flow chart schematic of this study is shown in Figure 5. This study is divided into three major parts: the calibration and validation of model parameters, the pre-processing of SM data, and the experimental part of DA.

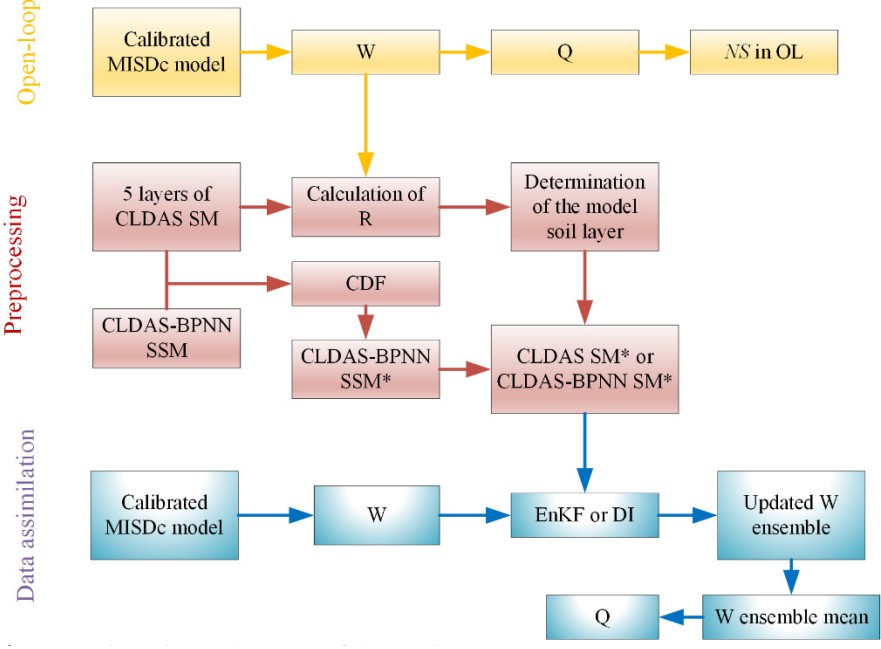

**Figure 5.** Flow chart schematic of the study.

## 3. Results

### 3.1. Model Calibration and Validation

Prior to DA, the MISDc model was driven using CLDAS rainfall and temperature data, and the observed discharge data were used to verify the accuracy of the simulated

runoff. From 1 January 2010 to 31 December 2014, the parameters were calibrated, and from 1 January 2015 to 31 December 2017, the model was validated. A comparison between the runoff simulated by the MISDc model and the observed discharge is shown in Figure 6. The used evaluation index formulas are shown in Equations (7)–(9), and their results and *R* are shown in Table 2.

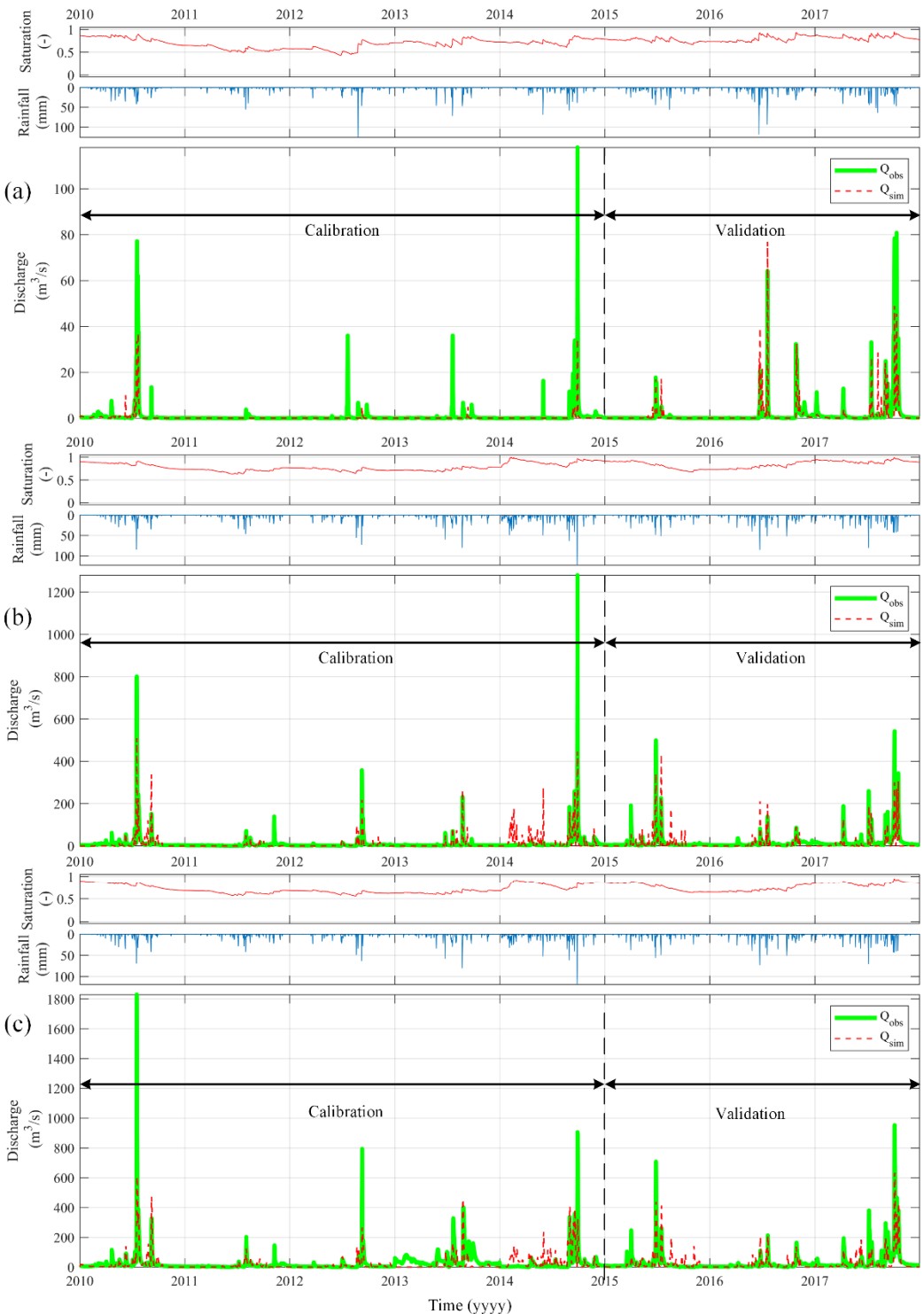

**Figure 6.** Comparisons of the observed, $Q_{obs}$, and simulated, $Q_{sim}$, discharge (lower panel) for the calibration and validation periods at the (**a**) WWQ, (**b**) DPL and (**c**) CTG basins. The upper panels also show the temporal pattern of soil saturation and rainfall.

**Table 2.** Performance of the MISDc model in the calibration and validation periods for the three basins.

| Catchments | Calibration (1 January 2010–31 December 2014) | | | Validation (1 January 2015–31 December 2017) | | |
|---|---|---|---|---|---|---|
| | *NS* | *ANSE* | *R* | *NS* | *ANSE* | *R* |
| WWQ | 0.494 | 0.532 | 0.799 | 0.512 | 0.688 | 0.737 |
| DPL | 0.561 | 0.641 | 0.750 | 0.594 | 0.806 | 0.795 |
| CTG | 0.374 | 0.487 | 0.676 | 0.628 | 0.856 | 0.829 |

As shown in Table 2, the *NS* coefficients of the three basins ranged from 0.374 to 0.561 during the calibration period and from 0.512 to 0.628 during the validation period. In general, the *NS* values in these three basins were generally accepted during the calibration and validation periods. Similarly, for the *ANSE* index, values in the calibration period ranged from 0.487 to 0.532, and values in the validation period ranged from 0.688 to 0.856. These results may be due to the fact that the model did not simulate high flows well in all three basins during the calibration period, while the model was able to simulate the flooding process better during the validation period (Figure 6). In general, the results of the *ANSE* values were satisfactory. As for *R*, the values were greater than 0.65 in the three basins, in both the calibration and validation periods. In general, the results of linear fitting were acceptable.

Overall, the simulation results of the model were not perfect, probably due to errors in our input rainfall and temperature data. Still, such results were satisfactory enough for our DA experiment. Moreover, the results of the model in the open-loop (OL, i.e., without assimilation) for the period of the DA experiment (31 March 2015–31 December 2017) were good enough.

### 3.2. The Influence of Different Numbers of Ensemble Members on the Results of the EnKF Method

In order to satisfy the problems of both computational accuracy and computational cost, the number of EnKF ensemble members was investigated in this study. We calculated the *NRMSE* values by varying the number of ensemble members (20, 50, 100, 200, and 500) during the assimilation period (31 March 2015–31 December 2017) and determining the size of the number of ensemble members by comparing the *NRMSE* values. Figure 7 shows the *NRMSE* values for these three basins.

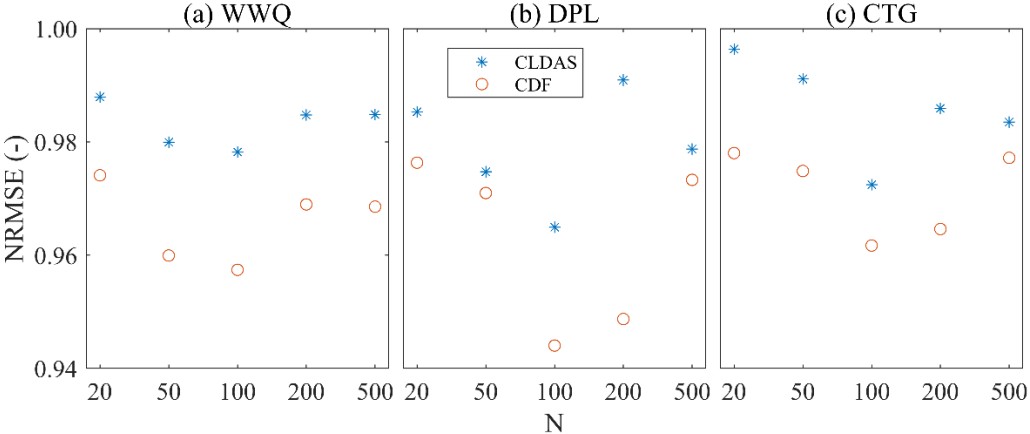

**Figure 7.** Variation of *NRMSE* with ensemble numbers of EnKF. (**a**) WWQ basin; (**b**) DPL basin; (**c**) CTG basin.

As we can see from Figure 7, when changing the number of ensemble members, the values of *NRMSE* of the three basins did not numerically change much, fluctuating between 0.97–0.99 of CLDAS SM DA and 0.95–0.98 of BPNN SM DA for the WWQ basin,

between 0.96–0.99 of CLDAS SM DA and 0.94–0.98 of BPNN SM DA for the DPL basin, and between 0.97–0.99 of CLDAS SM DA and 0.96–0.98 of BPNN SM DA for the CTG basin. The growth rates of *NS* calculated for different numbers of ensemble members were counted, and the results are shown in Figure 8. As can be seen from Figure 8, no outliers were found in the growth rate of *NS* calculated by changing the size of the ensemble members, and they all fluctuated within a normal range. Through the boxplot, we can conclude that changing the size of the ensemble members had little influence on the result of the EnKF SM DA.

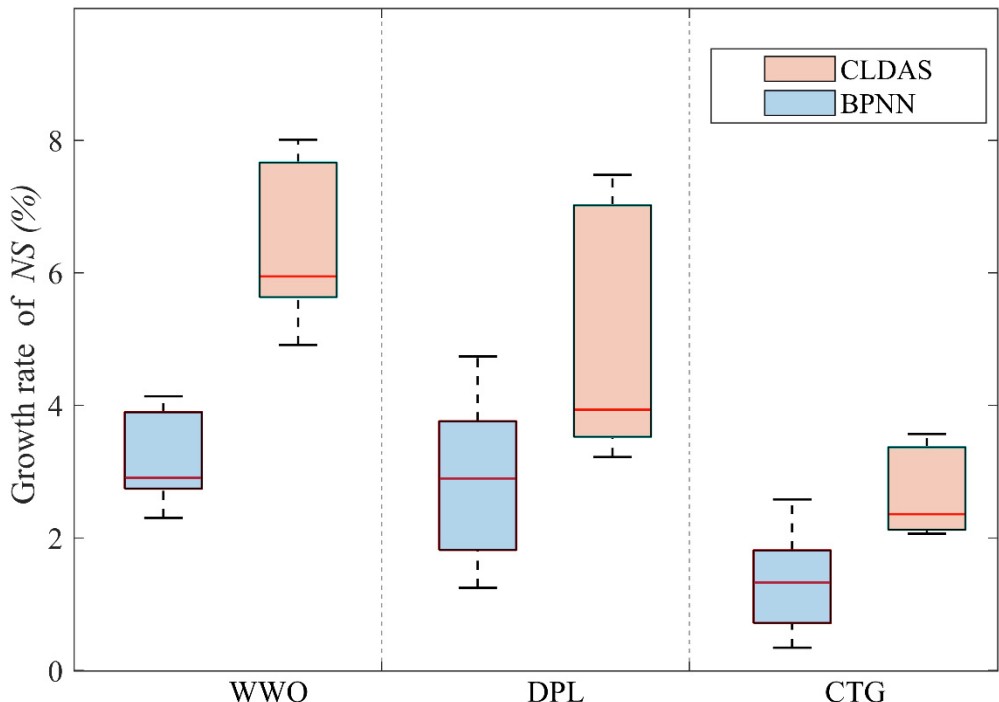

**Figure 8.** Boxplot of the growth rate of the *NS* for the three basins.

However, for all three basins, when the number of ensemble members was set to 100, the values of *NRMSE* were all minimum. Therefore, when conducting the EnKF DA experiment, the size of *N* was decided to be set to 100. In addition, as can be seen from Figures 7 and 8, when BPNN SM DA was carried out, the indicators of runoff results obtained by the model simulation were generally superior to those obtained by CLDAS SM DA.

*3.3. Data Assimilation Experiments*

In this paper, two DA methods were used, i.e., the direct insertion (DI) method and the ensemble Kalman filter (EnKF) method. Two types of SM data were assimilated into the MISDc model to compare the effect of assimilating the corrected SM data (BPNN SM) and the uncorrected SM data (CLDAS SM) on the model runoff simulation results. The study period was from 31 March 2015 to 31 December 2017. The results of the *NS* coefficients are represented in the "OL" row of the Table 3, where "OL" indicates the open-loop state or only the result of model simulation without assimilation.

**Table 3.** Results of the two methods SM DA for the three basins.

| Catchments | | WWQ | | DPL | | CTG | |
|---|---|---|---|---|---|---|---|
| | | *NS* | *NRMSE* | *NS* | *NRMSE* | *NS* | *NRMSE* |
| OL | | 0.510 | | 0.593 | | 0.678 | |
| DI | CLDAS | 0.025 | 1.411 | −0.121 | 1.660 | 0.120 | 1.653 |
| | BPNN | 0.007 | 1.423 | −0.124 | 1.662 | 0.109 | 1.664 |
| EnKF | CLDAS | 0.531 | 0.978 | 0.621 | 0.965 | 0.696 | 0.972 |
| | BPNN | 0.551 | 0.957 | 0.637 | 0.944 | 0.702 | 0.962 |

Using the DI method, the SM content calculated by the model ($SM_{sim}$) was directly replaced by the observed SM data ($SM_{obs}$). In the flowchart in Figure 4, the DI method is indicated by the blue dotted line. Here, the DI method did not involve the SWB model component of the MISDc model. Instead, the simple linear relationship equation was directly calculated using $SM_{obs}$ data, and then the result was brought into the RR model component to obtain the assimilated flow ($Q_a$). The results of the two kinds of SM data assimilated into the MISDc model using this method are shown in the "DI" row of Table 3. From the results, we can see that when the DI method was used, the values of the *NS* coefficients in all three basins decreased. The values of *NRMSE* were greater than 1 (indicating worsening results), regardless of which of the two kinds of SM data were assimilated. This phenomenon indicated that the results were less satisfactory when only the model-simulated SM data were replaced by the soil moisture data for the DA experiment, i.e., when the DI method was used. This is a well-known possibility in land DA. The DI method assumes the observations are perfect but ignores the influence of the structure of the hydrological model on the errors caused by real-world generalization. It has been shown the overestimating the quality of assimilated observations can degrade relative a DA system relative to an open loop. Therefore, there was a strong need for an experimental DA study using the EnKF method.

The EnKF method was implemented as described in Section 2.7. In the flowchart in Figure 4, the EnKF method is indicated by the red dotted line (note that the time state process for the EnKF method is not shown in the figure). According to Section 2.7 and Figure 4, we know that the EnKF method considers not only the SM data used for assimilation ($SM_{obs}$) but also the SM data obtained from model calculations when performing SM DA ($SM_{obs}$ represents the optimal SM value obtained by the EnKF). This is how the EnKF method differs from the DI method. The issue of the parameters to be set when the EnKF method was used is described in Section 3.2. The "EnKF" row of Table 3 shows the results of the method assimilating the two kinds of SM data.

As we can see from Table 3, the results improved (all *NS* values increased and all *NRMSE* values were less than 1) when assimilating the two kinds of SM data into the MISDc model by using the EnKF method for each of the three basins. Among them, the most significant improvement in the results of model runoff simulations was found in the DPL basin (*NRMSE* = 0.965 and 0.944), followed by the WWQ basin (*NRMSE* = 0.978 and 0.957) and the CTG basin (*NRMSE* = 0.972 and 0.962). Additionally, when assimilating the SM data from the BPNN, i.e., SM data after deviation correction, the results of the performance indexes were generally better than the results of the performance indexes calculated by directly assimilating CLDAS SM. However, the improvement was not numerically very high.

### 3.4. The Impact of the EnKF Method on High and Low Flows

Although it can be concluded from Table 3 that the model runoff simulation results were improved when assimilating the two kinds of SM data into the MISDc model via the EnKF method, the table does not directly express improvements for high and low flows in these three basins. The selection rule was less than the 5th discharge percentile for the determination of low flows. To determine high flows, the selection rule had to be greater

than the 95th discharge percentile. Therefore, the high and low flows were analyzed within each of the three basins, and the results are summarized in Figure 9. The index of low flows was the difference between $NS_{(logQ)}$ before and after assimilation, $\Delta NS_{(logQ)}$, while the index of high flows was the difference of $ANSE$ before and after assimilation, $\Delta ANSE$. Therefore, values of $\Delta NS_{(logQ)}$ and $\Delta ANSE$ of greater than zero indicate the improvement of the assimilation results, while values of less than zero indicate the deterioration of the assimilation results.

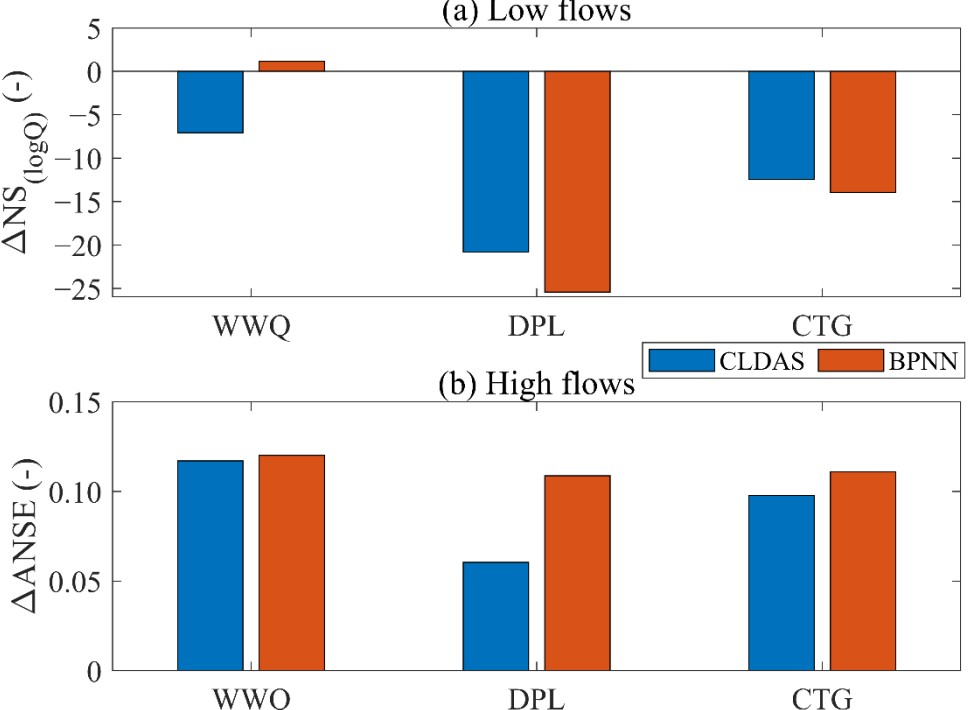

**Figure 9.** Differences ($\Delta NS_{(logQ)}$, and $\Delta ANSE$) between $NS_{(logQ)}$ and $ANSE$ performance scores were obtained before and after the assimilation of soil moisture for the three selected basins. While (**a**) $NS_{(logQ)}$ was well suited for highlighting the performance in the reproduction of low flows, (**b**) $ANSE$ was used for characterizing the agreement of high flows.

As can be seen from Figure 9, the EnKF method had a certain deterioration effect on low-flow simulations in all three basins (except for assimilation BPNN SM in the WWQ basin) demonstrated some improvements in high-flow simulations in all three basins. For our flood forecasting, we tend to focus more on high-flow situations. Therefore, improving the high-flow simulation results was satisfactory for our study. In Figure 9b, we can see that the red part is higher than the blue part, which indicates that a DA study using the corrected SM data could better improve the runoff simulation results of the model.

Generally, when assimilating CLDAS or BPNN SM data into the MISDc model using the EnKF method, the model somewhat deteriorated for low flows but somewhat improved for high and total period flows. This finding was satisfactory for flood forecasting in our small- and medium-sized river basins.

## 4. Discussion

In our present study, we investigated the performance of the MISDc model on three small and medium-sized river basins in the upper Huai River basin. In general, the performance of the MISDc model for these three basins was found to be satisfactory, although the model did not simulate the high flow component very well. This may have been due to errors in the input forcing data and topographic conditions in these three basins, which also occur in runoff simulations when using other models (e.g., HBV model; see Appendix A). Therefore, there was an urgency for us to conduct a DA study. In the DA experiments, we

investigated the effect of assimilating CLDAS SM data with the DI and EnKF methods on the runoff simulation capability of the MISDc model. However, the results of the model runoff simulation deteriorated when the DI method was used, a finding that was similar to that of Nayak, et al. [44]. Nayak, et al. [44] used GLDAS SM as the initial SM condition of a model (that is, the DI method in this paper), but found that the performance of the model decreased. This demonstrates the importance of model errors in data assimilation techniques. Berardi, et al. [45] proposed a new data assimilation technique based on the EnKF that was applied in some comprehensive experiments showing improvements over the classical ensemble Kalman filter, especially for problems with large model errors. Jamal and Linker [46] proposed an adaptive inflation method within the EnKF framework that updates the inflation factor in each time step based on model predictions and collected measurements, and they conducted two case studies and found that the method outperformed other existing methods. These results could be helpful for our future work. In the EnKF method, the assimilation of SM data contributed to the improvement of the model runoff simulation, and the CLDAS SM DA led to improvement for high flows but deterioration for low flows.

For the small- and medium-sized river basins we studied, flash floods quickly come and go, so timely forecasting helps us to prevent and control disasters. Therefore, EnKF SM DA is particularly important for improving high-flow simulation. However, EnKF SM DA worsened the simulation of low flow in our study, which was similar to the results of Massari, et al. [33]. Massari, et al. [33] found that when SM was assimilated into the MISDc model, the simulation of low flow was deteriorated for most basins while the simulation of high flow was improved.

In this study, CLDAS SM was used in our DA experiment. Compared with remote sensing SM products, CLDAS SM has real-time data release, which is convenient for our future research on SM DA to improve the runoff prediction ability of the hydrological model. In fact, SMOS and SMAP satellite remote sensing SM products also have RZSM data (SMOS L4 and SMAP L4) [47,48], so in future studies, we can consider adding the RZSM data of these two products to compare them with CLDAS SM products. When conducting the DA experiments, we did not update the model parameters regardless of whether the DI method or the EnKF method was used. In fact, whether or not a model's parameters are updated when conducting DA experiments was shown to make a difference in the final results [49]. Therefore, we suspect that this may be one of the reasons why SM DA using the DI method led to the decline of the runoff simulation ability of the model, and this aspect can be further researched in a follow-up study.

## 5. Conclusions

An experimental DA study of the MISDc model was conducted on three small- and medium-sized river basins in the upper Huai River basin using the DI and EnKF methods. The CLDAS SM was assimilated into BPNN SM data after the correction of CLDAS using a back propagation neural network and cumulative distribution function to improve the hydrological model's runoff simulation. The MISDc model's runoff simulation deteriorated when the DI method was used, regardless of the assimilated SM data. However, when the EnKF method was used, the ability to assimilate SM data to simulate runoff from the MISDc model improved and the improvement was more pronounced when assimilating BPNN SM data. Although the EnKF method showed deterioration in the results for model simulations at low flows, its improvement for model simulations at high flows cannot be ignored. While varying the number of ensemble members for the EnKF method, the fluctuation of the *NRMSE* was not very large. Still, the value of the *NRMSE* reached its minimum when the number of ensemble members was 100 in all three study areas. This finding led us to set the number of ensemble members to 100 when using the EnKF in this study, which ensured the accuracy of the calculation results and reduced the cost of the calculation. Since CLDAS SM data are updated in real time compared with remote sensing SM data, and the accuracy of CLDAS SM data is higher than that of other similar products

in China, assimilating CLDAS SM data into hydrological models to improve flash flood simulation is a feasible method.

**Author Contributions:** Conceptualization, Z.D.; Data curation, Z.D., H.L., Y.Z., Q.G. and X.W.; Formal analysis, Z.D., Q.G., X.W. and E.L.; Funding acquisition, H.L. and Y.Z.; Investigation, Z.D.; Methodology, Z.D., H.L. and Y.Z.; Project administration, H.L.; Resources, Z.D., H.L. and Y.Z.; Software, Z.D., H.L. and M.S.; Supervision, H.L. and Q.G.; Validation, Z.D. and M.S.; Visualization, Z.D.; Writing—original draft, Z.D.; Writing—review & editing, Z.D., H.L., N. A., Y.Z., Q.G., X.W., E.L., H.X. and Y.P. All authors have read and agreed to the published version of the manuscript.

**Funding:** This research is supported by the National Key Research and Development Program (Grant Nos. 2019YFC1510504); National Natural Science Foundation of China (Grant Nos. 41830752, 42071033, and 41961134003).

**Acknowledgments:** We are very grateful to the reviewers for their constructive comments and thoughtful suggestions that have improved this paper substantially.

**Conflicts of Interest:** The authors declare no conflict of interest.

## Appendix A

We input the same rainfall and temperature data into the HBV model. The calculated indexes are shown in Table A1, and the flood process line is shown in Figure A1. By comparing Figure 6 with Figure A1, Tables 2 and A1, we found that the HBV model also did not well-simulate the process of high flow. The two models failed to well-simulate the high flow process in the three basins, which may have been due to input forcing data errors and topographic factors in the three basins. The specific reasons need to be further studied.

**Table A1.** Performance of the HBV model in the calibration and validation periods for the three basins.

| Catchments | Calibration (1 January 2010–31 December 2013) | | | Validation (1 January 2014–31 December 2016) | | |
|---|---|---|---|---|---|---|
| | *NS* | *ANSE* | *R* | *NS* | *ANSE* | *R* |
| WWQ | 0.287 | 0.321 | 0.536 | 0.357 | 0.442 | 0.613 |
| DPL | 0.464 | 0.586 | 0.687 | 0.727 | 0.760 | 0.865 |
| CTG | 0.471 | 0.566 | 0.689 | 0.610 | 0.741 | 0.781 |

**Table A2.** Full name of the abbreviated nouns used in the study.

| Abbreviation | Full Name | Abbreviation | Full Name | Abbreviation | Full Name |
|---|---|---|---|---|---|
| CDF | Cumulative distribution function | DI | Direct insertion | MISDc | Modello Idrologico Semi-Distribuito in continuo |
| CLDAS | China land data assimilation system | DPL | Dapoling | | |
| CTG | Changtaiguan | EnKF | Ensemble Kalman filter | SM | Soil moisture |
| DA | Data assimilation | KF | Kalman filter | WWQ | Wangwuqiao |

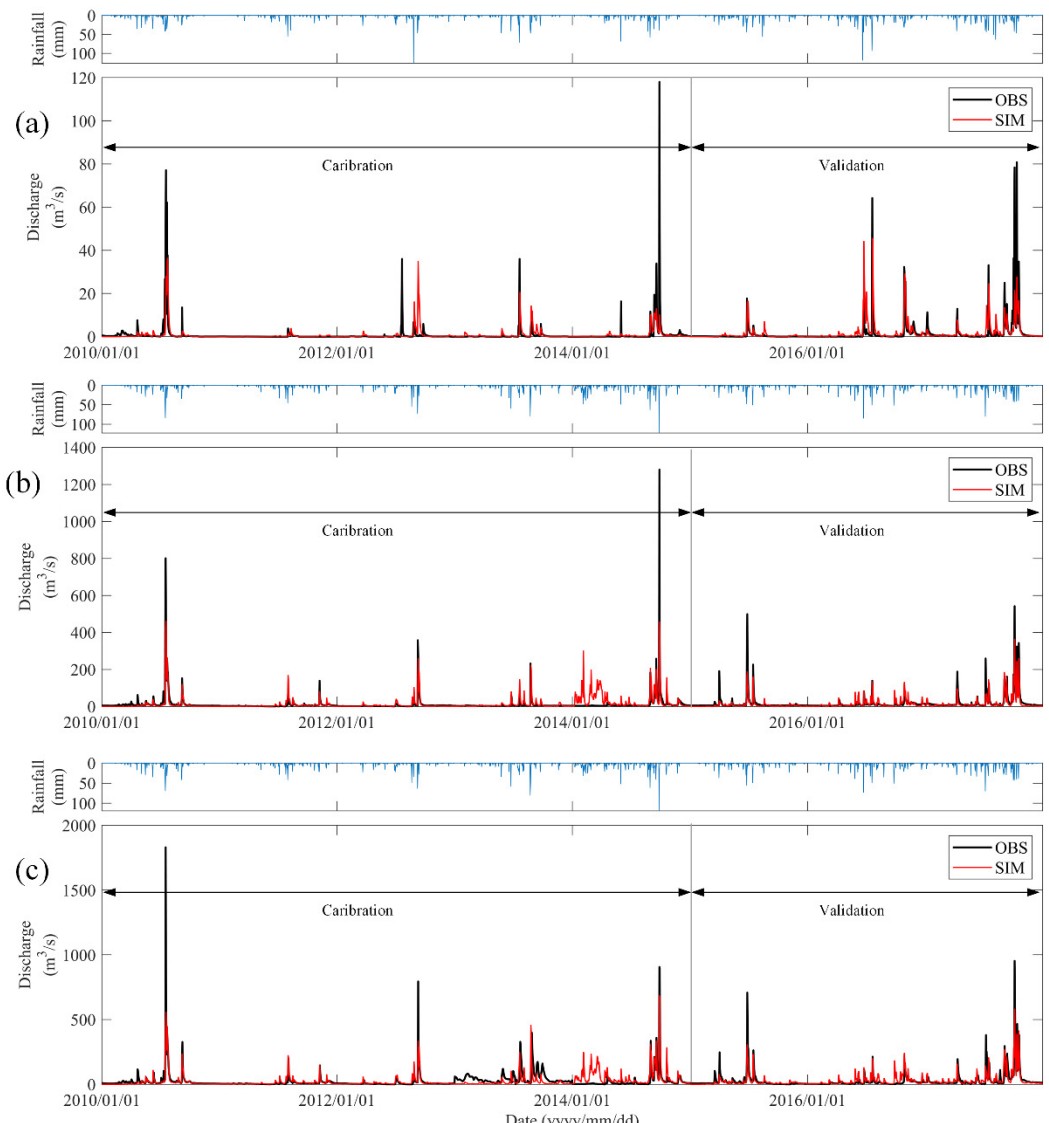

**Figure A1.** Comparisons of the observed, $Q_{obs}$, and simulated, $Q_{sim}$, discharge (lower panel) for the calibration and validation periods by the HBV model at the (**a**) WWQ, (**b**) DPL and (**c**) CTG basins.

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
