# Peer review of "Soil Moisture Data Assimilation in MISDc for Improved Hydrological Simulation in Upper Huai River Basin, China"

_water, doi:10.3390/w14213476_

Round 1

Reviewer 2 Report

The paper Soil Moisture Data Assimilation in MISDc for Improved Hy-2 drological Simulation in Upper Huai River Basin, China  presents an interesting study concearning improving the runoff forecasting from hydrological simulations with MISDc model during flash floods, using remotely sensed data on soil moisture.

The paper is well written and structured, English language and style are good. The objectives are formulated, methods are clearly explained, results are discussed objectively and conclusions are drawn based on results.

The paper has 19 pages of which about 3 for the Introduction, 6 pages for the Materials and methods, 6.5 pages for Results and discussions, 0.5 for Conclusions and 2 for References. Citations are appropriate, in the field of the paper, most of them from highly referenced journals or conferences.

However, authors should consider citing another 2 very similar papers in the field and area:

ESA CCI Soil Moisture Assimilation in SWAT for Improved Hydrological Simulation in Upper Huai River Basin, Advances in Meteorology, 2018

Influence of Calibration Parameter Selection on Flash Flood Simulation for Small to Medium Catchments with MISDc-2L Model, Water, 2018

Another advice for the authors would be to consider adding a list of acronyms in an Annex at the end of the paper, which would be useful for the readers. I consider authors use too many acronyms in the text, some of them not explained before, which make the understanding a bit difficult.

Beneath are some examples of formal issues found in the text.

·         Row 17, 83 Please explain the acronyms (MISDc, ESA CCI) first time you use them in text. Overall is rather difficult to read the text due to multiple accronyms. One has always to look back to remember... Try to reduce their number, especially in the Abstract and Conclusions.

·         R. 18 assimilated

·         R 25 Data not in bold

·         R 53-58. Try to split the sentence.

·         R 62-65 Please review sentence

·         R 95-97 lacks predicate

·         R 144 BPNN SSM

·         R 191 basins

·         R 257 (The t)wo models

·         R 335 A priori or prior

In conclusion, I recommend publishing the paper with aforementioned observations.

Author Response

Point 1: However, authors should consider citing another 2 very similar papers in the field and area:

ESA CCI Soil Moisture Assimilation in SWAT for Improved Hydrological Simulation in Upper Huai River Basin, Advances in Meteorology, 2018

Influence of Calibration Parameter Selection on Flash Flood Simulation for Small to Medium Catchments with MISDc-2L Model, Water, 2018

 Response 1: Thanks for the reviewer's comments. We have made references to both papers in the manuscript.

Point 2: Another advice for the authors would be to consider adding a list of acronyms in an Annex at the end of the paper, which would be useful for the readers. I consider authors use too many acronyms in the text, some of them not explained before, which make the understanding a bit difficult.

Response 2: We thank the reviewers for their suggestions. We have added a part on abbreviations in the appendix section.

Point 3: Beneath are some examples of formal issues found in the text.

  • Row 17, 83 Please explain the acronyms (MISDc, ESA CCI) first time you use them in text. Overall is rather difficult to read the text due to multiple accronyms. One has always to look back to remember... Try to reduce their number, especially in the Abstract and Conclusions.
  • R. 18 assimilated
  • R 25 Data not in bold
  • R 53-58. Try to split the sentence.
  • R 62-65 Please review sentence
  • R 95-97 lacks predicate
  • R 144 BPNN SSM
  • R 191 basins
  • R 257 (The t)wo models
  • R 335 A priori or prior

Response 3: Regarding these issues of English grammatical structure, we will get a native English speaker to help with the revisions.

Reviewer 3 Report

Authors make use of CLDAS SM data product published by the China Meteorological Administration, and make use of "Modello Idrologico Semi-Distribuito in continuo’ (MISDc) model. I appreciate the idea of making use of a well-established European model into a Chinese application framework. The paper is well written. Nevertheless, before being considered for publication, I think some issues need to be faced, listed below.

-Authors acknowledge the importance of model errors in Data Assimilation techniques.  To this purpose, it could be useful to refer also, in the discussion,
 to some significant papers dealing with model errors in hydrological applications, such as

Berardi et al, CPC 2016, https://doi.org/10.1016/j.cpc.2016.07.025
 (in which soil moisture states are corrected by a novel DA technique) or

Jamal and Linker VZJ 2020 https://doi.org/10.1002/vzj2.20000.
- Line 95-98: the verb seems to be missing.
- A definition of "lumped hydrological model" could be useful for the reader.
- Line 229. I guess "China" and not "Chia".
- A question about CLDAS SM product. Is it freely available? If so, it is worth to specify it in the manuscript.
- Line 257. What does "wo models" mean?
- Line 261. Where, the water content balance equation is the basis, and infiltration is expressed by Green-Ampt equation. Please rephrase this sentence.
- In Section 2.4, I would recap the MISDc model equations.
- In Equation (3), authors briefly recap EnKF method, which is fine. However, I think they should elaborate a little bit, in the following, on how to apply EnKF in the case at hand. This comment is strictly related to the previous one.

Author Response

Point 1: Authors acknowledge the importance of model errors in Data Assimilation techniques. To this purpose, it could be useful to refer also, in the discussion, to some significant papers dealing with model errors in hydrological applications, such as Berardi et al, CPC 2016, https://doi.org/10.1016/j.cpc.2016.07.025 (in which soil moisture states are corrected by a novel DA technique) or Jamal and Linker VZJ 2020 https://doi.org/10.1002/vzj2.20000.

Response 1: Many thanks to the reviewers for their comments and recommended relevant papers, which were very helpful and we have added them to the discussion section.

Point 2: Line 95-98: the verb seems to be missing.

Response 2: Ok, we revised the sentence and will get a native English speaker to help with the revisions.

Point 3: A definition of "lumped hydrological model" could be useful for the reader.

Response 3: OK, we added the definition of "lumped hydrological model" where it first appears.

Point 4: Line 229. I guess "China" and not "Chia".

Response 4: Yes, it should be "China".

Point 5: A question about CLDAS SM product. Is it freely available? If so, it is worth to specify it in the manuscript.

Response 5: Yes, the CLDAS data is free and open source, and we provide the URL information in the sixth paragraph of the introduction section, which the reader can access to search and download.

Point 6: Line 257. What does "wo models" mean?

Response 6: It has been modified, "wo models" has been changed to "Two models".

Point 7: Line 261. Where, the water content balance equation is the basis, and infiltration is expressed by Green-Ampt equation. Please rephrase this sentence.

Response 7: We have revised the sentence and will get a native English speaker to help with the revisions.

Point 8: In Section 2.4, I would recap the MISDc model equations.

Response 8: The MISDc model has too many equations, too much detail for the length of the manuscript, and our lack of detail may cause confusion to the readers. Therefore, we provide two related articles that describe the structure of the MISDc model in detail, which we think will be more helpful to the readers, if they are interested in the MISDc model.

Point 9: In Equation (3), authors briefly recap EnKF method, which is fine. However, I think they should elaborate a little bit, in the following, on how to apply EnKF in the case at hand. This comment is strictly related to the previous one.

Response 9: Figure 4 of the latest version of the manuscript shows an example of our application of the EnKF method or the DI method in our model. Also, we added Figure 5 as a flow chart schematic of our study. Since we have described the EnKF calculation method in detail in Section EnKF, we do not elaborate on the Figure 4.

Round 2

Reviewer 1 Report

Accept

Author Response

Thank you!

Reviewer 3 Report

The paper has been improved.

To be honest, I have to notice that the reply to the first comment is not correct:

Response 1: Many thanks to the reviewers for their comments and recommended relevant papers, which were very helpful and we have added them to the discussion section.

Indeed, I have to say that this is not true. The paper by Jamal and Linker has not been cited, and the paper Berardi et al is cited properly cited, but in a wrong way (the paper is dated 2016  and not 2015, and the journal has not been cited, differently from other citations). PLEASE CORRECT

Author Response

OK, we have revised that section. The counterpart of the manuscript reads (Lines 522 to 528 of the manuscript) :

Berardi et al. (2016) proposed a new data assimilation technique based on the EnKF, which was applied in some comprehensive experiments showing improvements over the classical ensemble Kalman filter, especially for problems with large model errors. Jamal and Linker (2020)proposed an adaptive inflation method within the EnKF framework that updates the inflation factor in each time step based on model predictions and collected measurements, and conducted two case studies and found that the method outperforms other existing methods.